# NEAR-OPTIMAL GLIMPSE SEQUENCES FOR IMPROVED HARD ATTENTION NEURAL NETWORK TRAINING

## ABSTRACT

Hard visual attention is a promising approach to reduce the computational burden of modern computer vision methodologies. Hard attention mechanisms are typically non-differentiable. They can be trained with reinforcement learning but the high-variance training this entails hinders more widespread application. We show how hard attention for image classification can be framed as a Bayesian optimal experimental design (BOED) problem. From this perspective, the optimal locations to attend to are those which provide the greatest expected reduction in the entropy of the classification distribution. We introduce methodology from the BOED literature to approximate this optimal behaviour, and use it to generate 'near-optimal' sequences of attention locations. We then show how to use such sequences to partially supervise, and therefore speed up, the training of a hard attention mechanism. Although generating these sequences is computationally expensive, they can be reused by any other networks later trained on the same task.

## 1 INTRODUCTION

Attention can be defined as the "allocation of limited cognitive processing resources" (Anderson, 2005). In humans the density of photoreceptors varies across the retina. It is much greater in the centre (Bear et al., 2007) and covers an approximately 210 degree field of view (Traquair, 1949). This means that the visual system is a limited resource with respect to observing the environment and that it must be allocated, or controlled, by some attention mechanism. We refer to this kind of controlled allocation of limited sensor resources as "hard" attention. This is in contrast with "soft" attention, the controlled application of limited computational resources to full sensory input. Hard attention can solve certain tasks using orders of magnitude less sensor bandwidth and computation than the alternatives (Katharopoulos & Fleuret, 2019; Rensink, 2000). It therefore may enable the use of modern approaches to computer vision in low-power settings such as mobile devices.

This paper focuses on the application of hard attention in image classification. Our model of attention (shown in Fig. 1) is as follows: a recurrent neural network (RNN) is given $T$ steps to classify some unchanging input image. Before each step, the RNN outputs the coordinates of a pixel in the image. A patch of the image centered around this pixel is then fed into the RNN. We call this image patch a glimpse, and the coordinates a glimpse location. As such, the RNN controls its input by selecting each glimpse location, and this decision can be based on previous glimpses. After $T$ steps, the RNN's hidden state is mapped to a classification output. As with most artificial hard attention mechanisms (Mnih et al., 2014; Ba et al., 2014), this output is not differentiable with respect to the sequence of glimpse locations selected. This makes training with standard gradient backpropagation impossible, and so high variance gradient estimators such as REINFORCE (Williams, 1992) are commonly used instead (Mnih et al., 2014; Ba et al., 2014). The resulting noisy gradient estimates make training difficult, especially for large $T$.

In order to improve hard attention training, we take inspiration from neuroscience literature which suggests that visual attention is directed so as to maximally reduce entropy in an agent's world model (Bruce & Tsotsos, 2009; Itti & Baldi, 2009; Schwartenbeck et al., 2013; Feldman & Friston, 2010). There is a corresponding mathematical formulation of such an objective, namely Bayesian optimal experimental design (BOED) (Chaloner & Verdinelli, 1995). BOED tackles the problem

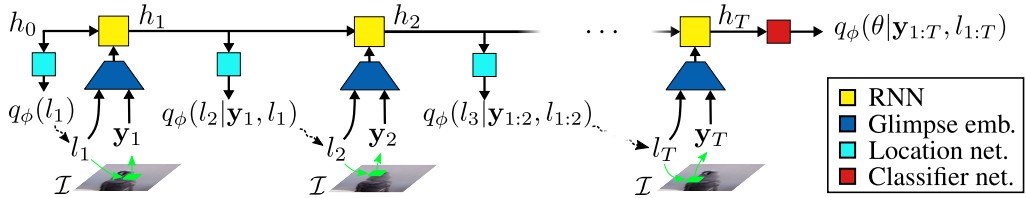

Figure 1: The hard attention network architecture we consider, consisting of an RNN core (yellow), a location network (light blue), a glimpse embedder (dark blue), and a classifier (red). $h_t$ is the RNN hidden state after $t$ steps. The network outputs distributions over where to attend ($l_t$) at each time step, and over the class label ($\theta$) after $T$ steps.

of designing an experiment to maximally reduce uncertainty in some unknown variable. When classifying an image with hard visual attention, the 'experiment' is the process of taking a glimpse; the 'design' is the glimpse location; and the unknown variable is the class label. In general, BOED is applicable only when a probabilistic model of the experiment exists. This could be, for example, a prior distribution over the class label and a generative model for the observed image patch conditioned on the class label and glimpse location. We leverage generative adversarial networks (GANs) (Goodfellow et al., 2014) to provide such a model.

We use methodology from BOED to introduce the following training procedure for hard attention networks, which we call partial supervision by near-optimal glimpse sequences (PS-NOGS).

1. We assume that we are given an image classification task and a corresponding labelled dataset. Then, for a subset of the training images, we determine an approximately optimal (in the BOED sense) glimpse location for a hard attention network to attend to at each time step. We refer to the resulting sequences of glimpse locations as near-optimal glimpse sequences. Section 4 describes our novel method to generate them.
2. We use these near-optimal glimpse sequences as an additional supervision signal for training a hard attention network. Section 5 introduces our novel training objective for this.

We empirically investigate the performance of PS-NOGS and find that it leads to faster training than our baselines, and qualitatively different behaviour with competitive accuracy. We validate the use of BOED to generate glimpse sequences through comparisons with supervision both by hand-crafted glimpse sequences, and by glimpse sequences sampled from a trained hard attention network.

## 2   HARD ATTENTION

Given an image, $\mathcal{I}$, we consider the task of inferring its label, $\theta$. We use an architecture based on that of Mnih et al. (2014), shown in Fig. 1. It runs for a fixed number of steps, $T$. At each step $t$, the RNN samples a glimpse location, $l_t$, from a distribution conditioned on previous glimpses via the RNN's hidden state. A glimpse, in the form of a contiguous square of pixels, is extracted from the image at this location. We denote this $\mathbf{y}_t = f_{\text{fovea}}(\mathcal{I}, l_t)$. An embedding of $\mathbf{y}_t$ and $l_t$ is then input to the RNN. After $T$ glimpses, the network outputs a classification distribution $q_\phi(\theta | \mathbf{y}_{1:T}, l_{1:T})$, where $\phi$ are the learnable network parameters. Mnih et al. (2014) use glimpses consisting of three image patches at different resolutions, but the architectures are otherwise identical. As it directly processes only a fraction of an image, this architecture is suited to low-power scenarios such as use on mobile devices.

During optimisation, gradients cannot be computed by simple backpropagation since $f_{\text{fovea}}$ is non-differentiable. An alternative, taken by Mnih et al. (2014) and others in the literature (Ba et al., 2014; Sermanet et al., 2014), is to obtain high-variance gradient estimates using REINFORCE (Williams, 1992). Although these are unbiased, their high variance has made scaling beyond simple problems such as digit classification (Netzer et al., 2011) challenging. Section 7 describes alternatives (Ba et al., 2015; Lawson et al., 2018) to training with REINFORCE, but similar problems with scalability exist. This has led many studies to focus on easing the learning task by altering the architecture: e.g., to process a downsampled image before selecting glimpse locations (Ba et al., 2014; Sermanet et al., 2014; Katharopoulos & Fleuret, 2019). We summarise these innovations in Section 7 but they tend to be less suitable for low-power computation. We therefore believe that improved training of the architecture in Fig. 1 is an important research problem, and it is the focus of this paper.

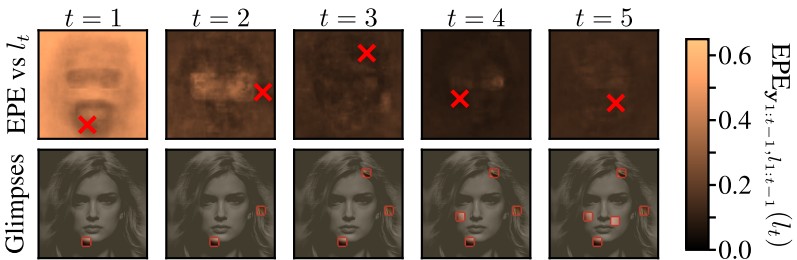

Figure 2: A near-optimal glimpse sequence being generated for the task of inferring the attribute 'Male'. **Top row:** A heatmap of estimated expected posterior entropy for each possible next glimpse location $l_t$. The red cross marks the minimum, which is chosen as the next glimpse location. **Bottom row:** Observed parts of the image after taking each glimpse.

## 3  BAYESIAN OPTIMAL EXPERIMENTAL DESIGN

Designing an experiment to be maximally informative is a fundamental problem that applies as much to tuning the parameters of a political survey (Warwick & Lininger, 1975) as to deciding where to direct attention to answer a query. BOED (Chaloner & Verdinelli, 1995) provides a unifying framework for this by allowing a formal comparison of possible experiments under problem-specific prior knowledge. Consider selecting the design, $l$, of an experiment to infer some unknown parameter, $\theta$. For example, $\theta$ may be the median lethal dose of a drug, and $l$ the doses of this drug given to various groups of rats (Chaloner & Verdinelli, 1995). Alternatively, as we consider in this paper, $\theta$ is the class label of an image and $l$ determines which part of the image we observe. The experiment results in a measurement of $\mathbf{y} \sim p(\mathbf{y}|l, \theta)$. Following the previous examples, $\mathbf{y}$ could be the number of rats which die in each group or the observed pixel values. Given a prior distribution over $\theta$ and knowledge of $p(\mathbf{y}|l, \theta)$, we can use the measurement to infer a posterior distribution over $\theta$ using Bayes' rule: $p(\theta|\mathbf{y}, l) = \frac{p(\mathbf{y}|l,\theta)p(\theta)}{\int p(\mathbf{y}|l,\theta)p(\theta)\mathrm{d}\theta}$. The aim of our experiment is to infer $\theta$, and so a well designed experiment will reduce the uncertainty about $\theta$ by as much as possible. The uncertainty after the experiment can be quantified by the Shannon entropy in the posterior,

$$\mathcal{H}\left[p(\theta|\mathbf{y}, l)\right] = \mathbb{E}_{p(\theta|\mathbf{y}, l)}\left[-\log p(\theta|\mathbf{y}, l)\right]. \tag{1}$$

To maximally reduce the uncertainty, we wish to select $l$ to minimise this posterior entropy. However, the design of the experiment must be chosen before $\mathbf{y}$ is measured and so we cannot evaluate the posterior entropy exactly. Instead, we minimise an expectation of it over $p(\mathbf{y}|l) = \mathbb{E}_{p(\theta)}\left[p(\mathbf{y}|l, \theta)\right]$, the marginal distribution of $\mathbf{y}$. This is the expected posterior entropy, or EPE.

$$\mathrm{EPE}(l) = \mathbb{E}_{p(\mathbf{y}|l)}\left[\mathcal{H}\left[p(\theta|\mathbf{y}, l)\right]\right]. \tag{2}$$

Above, we considered the case of selecting a one-off design for an experiment, such as taking a single glimpse. For the case where a sequence of glimpses can be taken, we need *sequential* experimental design. In this scenario, the choice of design $l_t$ can be informed by the designs and outcomes of previous experiments, $l_{1:t-1}$ and $\mathbf{y}_{1:t-1}$. The marginal distribution over outcomes is therefore $p(\mathbf{y}_t|l_{1:t}, \mathbf{y}_{1:t-1})$ rather than $p(\mathbf{y}_t|l_t)$. Similarly, the posterior after observing $\mathbf{y}_t$ is $p(\theta|l_{1:t}, \mathbf{y}_{1:t})$. Therefore, in the sequential case which we consider throughout the rest of the paper, we greedily minimise the following form of the EPE on each iteration:

$$\mathrm{EPE}_{\mathbf{y}_{1:t-1}, l_{1:t-1}}(l_t) = \mathbb{E}_{p(\mathbf{y}_t|\mathbf{y}_{1:t-1}, l_{1:t})}\left[\mathcal{H}\left[p(\theta|\mathbf{y}_{1:t}, l_{1:t})\right]\right]. \tag{3}$$

To summarise, sequential BOED involves, at each time $t$, selecting $l_t = \arg\min_{l_t} \mathrm{EPE}_{\mathbf{y}_{1:t-1}, l_{1:t-1}}(l_t)$ and then performing the experiment with design $l_t$ to observe $\mathbf{y}_t$.

## 4  GENERATING NEAR-OPTIMAL GLIMPSE SEQUENCES

**Role of BOED pipeline**   To reiterate the outline of our method, we first annotate a portion of the training data with glimpse sequences, and then in the second stage use these to speed up the training of a hard attention mechanism. This section details our BOED pipeline for the first stage.

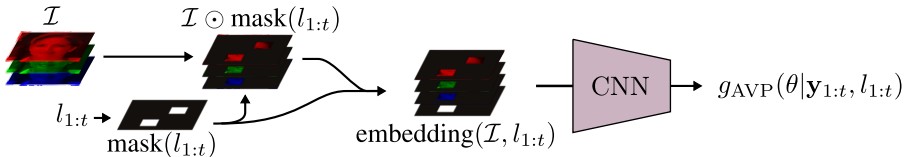

Figure 3: Attentional variational posterior CNN. An RGB image and $l_{1:t}$ are processed to create an embedding of the information gained from glimpses $1$ to $t$. This embedding is fed into an image classifier to obtain an approximation of $p(\theta|\mathbf{y}_{1:t}, l_{1:t})$.

**EPE estimator** BOED requires a probabilistic model of the measurements and parameters we wish to infer. That is, we need to define $p(\theta, \mathbf{y}_{1:t}|l_{1:t})$ for any $l_{1:t}$. To do so in the visual attention setting, we first define $p(\theta, \mathcal{I})$ to be the intractable joint distribution over labels and images from which our training and test data originate. To be consistent with our definition in Section 2 of $\mathbf{y}$ as a deterministic function of $\mathcal{I}$ and $l$, we then define $p(\mathbf{y}_i|\mathcal{I}, l_i)$ to be a Dirac-delta distribution on $f_{\text{fovea}}(\mathcal{I}, l_i)$. The joint distribution is then

$$p(\theta, \mathbf{y}_{1:t}|l_{1:t}) = \int p(\theta, \mathcal{I}) \prod_{i=1}^{t} p(\mathbf{y}_i|\mathcal{I}, l_i) \mathrm{d}\mathcal{I}. \tag{4}$$

Given this joint distribution, $\text{EPE}_{\mathbf{y}_{1:t-1}, l_{1:t-1}}(l_t)$ is well defined but intractable in general. We therefore consider how to approximate it. To simplify our method for doing so, we first rearrange the expression given in Eq. (3) so that the expectation is over $\mathcal{I}$ rather than $\mathbf{y}_t$. Taking advantage of the fact that $\mathbf{y}_i$ is a deterministic function of $\mathcal{I}$ and $l_i$ allows it to be rewritten as follows (proof in the appendix). Defining $f_{\text{fovea}}(\mathcal{I}, l_{1:t}) = \{f_{\text{fovea}}(\mathcal{I}, l_1), \ldots, f_{\text{fovea}}(\mathcal{I}, l_t)\}$,

$$\text{EPE}_{\mathbf{y}_{1:t-1}, l_{1:t-1}}(l_t) = \mathbb{E}_{p(\mathcal{I}|\mathbf{y}_{1:t-1}, l_{1:t-1})} \left[ \mathcal{H} \left[ p(\theta|f_{\text{fovea}}(\mathcal{I}, l_{1:t}), l_{1:t}) \right] \right]. \tag{5}$$

Given this form of the expected posterior entropy, we can approximate it if we can leverage the dataset to obtain:

- a learned *attentional variational posterior*, $g_{\text{AVP}}(\theta|\mathbf{y}_{1:t}, l_{1:t}) \approx p(\theta|\mathbf{y}_{1:t}, l_{1:t})$,
- and *stochastic image completion* distribution $r_{\text{img}}(\mathcal{I}|\mathbf{y}_{1:t-1}, l_{1:t-1}) \approx p(\mathcal{I}|\mathbf{y}_{1:t-1}, l_{1:t-1})$.

We expand on the form of each of these approximations later in this section. First, combining them with Eq. (5) and using a Monte Carlo estimate of the expectation yields our estimator for the EPE:

$$\text{EPE}_{\mathbf{y}_{1:t-1}, l_{1:t-1}}(l_t) \approx \frac{1}{N} \sum_{n=1}^{N} \mathcal{H} \left[ g_{\text{AVP}}(\theta|f_{\text{fovea}}(\mathcal{I}^{(n)}, l_{1:t}), l_{1:t}) \right] \tag{6}$$

with $\mathcal{I}^{(1)}, \ldots, \mathcal{I}^{(N)} \sim r_{\text{img}}(\mathcal{I}|\mathbf{y}_{1:t-1}, l_{1:t-1})$.

**Overview of BOED pipeline** We select $l_t$ with a grid search. That is, denoting the set of allowed values of $l_t$ as $L$, we compute our approximation of $\text{EPE}_{\mathbf{y}_{1:t-1}, l_{1:t-1}}(l_t)$ for all $l_t \in L$. We then select the value of $l_t$ for which this is least. To do so, our full BOED pipeline is as follows.

1. Sample $\mathcal{I}^{(1)} \ldots, \mathcal{I}^{(N)} \sim r_{\text{img}}(\mathcal{I}|\mathbf{y}_{1:t-1}, l_{1:t-1})$.
2. For each $l_t \in L$, approximate the expected posterior entropy with Eq. (6).
3. Select the value of $l_t$ for which this approximation is least.

Repeating these steps for $t = 1, \ldots, T$ yields a near-optimal glimpse sequence $l_{1:T}$ for image $\mathcal{I}$. Figure 2 shows an example of this process. We must do this for all images in some subset of a dataset to be able to partially supervise hard attention training as described in Section 5. We now describe the form of $g_{\text{AVP}}$ (the attentional variational posterior) and $r_{\text{img}}$ (stochastic image completion).

**Attentional variational posterior** In this section we introduce our novel approach for efficiently approximating the intractable posterior $p(\theta|\mathbf{y}_{1:t}, l_{1:t})$. We train a convolutional neural network (CNN) to map from a sequence of glimpses , $\mathbf{y}_{1:t}$, and their locations, $l_{1:t}$, to $g_{\text{AVP}}(\theta|\mathbf{y}_{1:t}, l_{1:t})$, an approximation of this posterior. We call this the attentional variational posterior CNN (AVP-CNN). To allow a single CNN to cope with varying $\mathbf{y}_{1:t}$, $l_{1:t}$, and even varying $t$, we embed its input as shown in Fig. 3. Essentially, $l_{1:t}$ is used to create an image-sized mask which is 1 for observed

pixels and 0 for unobserved pixels. Elementwise multiplication of this mask with the input image sets unobserved pixels to zero. The mask is then concatenated as an additional channel. This embedding naturally maintains spatial information while enforcing an invariance to permutations of the glimpse sequence. We use a Densenet-121 (Huang et al., 2017) CNN architecture (pretrained on ImageNet (Deng et al., 2009)) to map from this embedding to a vector of probabilities representing $g_{\text{AVP}}$. We train the network to minimise the KL divergence between its output and $p(\theta|\mathbf{y}_{1:t}, l_{1:t})$. That is, $D_{KL}(p(\theta|\mathbf{y}_{1:t}, l_{1:t})||g_{\text{AVP}}(\theta|\mathbf{y}_{1:t}, l_{1:t}))$. To ensure that $g_{\text{AVP}}$ is close for all $t$, $l_{1:t}$ and $\mathbf{y}_{1:t}$, the loss used is an expectation of this KL divergence over $p(\mathbf{y}_{1:t}|l_{1:t})u(t, l_{1:t})$. We factorise $u(t, l_{1:t})$ as $u(t) \prod_{i=1}^{t} u(l_i)$ where, so that all times and glimpse locations are weighted equally in the loss, $u(t)$ is a uniform distribution over $1, \dots, T$ and $u(l_i)$ is a uniform distribution over all image locations. Denoting the network parameters $\lambda$, the gradient of this loss is

$$\frac{\partial}{\partial \lambda} \mathcal{L}_\lambda = \mathbb{E}_{p(\theta, \mathbf{y}_{1:t}|l_{1:t})u(t, l_{1:t})} \left[ -\frac{\partial}{\partial \lambda} \log g_{\text{AVP}}^\lambda(\theta|\mathbf{y}_{1:t}, l_{1:t}) \right]. \tag{7}$$

This gradient is the same as that of a cross-entropy loss on data sampled from $p(\theta, \mathbf{y}_{1:t}|l_{1:t})u(t, l_{1:t})$, and can be approximated by a Monte Carlo estimate.

Our approximation of the EPE in Eq. (6) involves the entropy of $g_{\text{AVP}}$. Since $g_{\text{AVP}}$ is a categorical distribution, this is simply computed analytically. This amortised approximation of the posterior entropy is inspired by Foster et al. (2019), but has two important differences to their estimator:

- Foster et al. learn a mapping from $\mathbf{y}_t$ to $g(\theta|\mathbf{y}_{1:t}, l_{1:t})$, sharing information between "nearby" samples of $\mathbf{y}_t$ to reduce the computational cost of the experimental design. Our AVP-CNN takes this amortization further by learning a single mapping from $t$, $l_{1:t}$ and $\mathbf{y}_{1:t}$ to $g_{\text{AVP}}(\theta|\mathbf{y}_{1:t}, l_{1:t})$, which yields significant further efficiency gains in our setting.
- Whereas we approximate $\mathcal{H}[p]$ with $\mathcal{H}[g_{\text{AVP}}] = \mathbb{E}_{g_{\text{AVP}}}[-\log g_{\text{AVP}}]$, Foster et al. use $\mathbb{E}_p[-\log g]$. This provides an upper bound on $\mathcal{H}[p]$ but is not applicable in our case as we cannot sample from $p(\theta|\mathbf{y}_{1:t}, l_{1:t})$. Both approximations are exact when $g_{\text{AVP}} = p$.

**Stochastic image completion**  We considered numerous ways to form $r_{\text{img}}(\mathcal{I}|\mathbf{y}_{1:t-1}, l_{1:t-1})$ including inpainting (Pathak et al., 2016; Isola et al., 2017) and Markov chain Monte Carlo in a generative model. Future research in generative modelling may provide alternatives to this component of our method but, for now, we choose to represent $r_{\text{img}}$ using a technique we developed based on image retrieval (Jégou et al., 2010). Of the methods we considered, this gave the best trade-off between speed and sample quality. It involves creating an empirical image distribution with 1.5 million images for each experiment using GANs with publicly available pre-trained weights (StyleGAN (Karras et al., 2018) for CelebA-HQ and FineGAN (Singh et al., 2019) for Caltech-UCSD Birds). We note that the use of pre-trained models makes test leakage possible but verify in Appendix C.4 that this is unlikely to impact our results. During sampling, the database is searched for images that 'match' the previous glimpses ($\mathbf{y}_{1:t-1}$ and $l_{1:t-1}$). How well these glimpses match a database image, $\mathcal{I}'$, is measured by the squared distance in pixel space at glimpse locations: $\sum_{i=1}^{t-1} \|\mathbf{y}_i - f_{\text{fovea}}(\mathcal{I}', l_i)\|_2^2$. This distance defines a probability distribution over the images in the database. To reduce computation, we first compare approximations of the observed parts of each image using principal component analysis (Jolliffe, 2011), and compute exact distances only when these are close. The overall procedure to sample from $r_{\text{img}}$ corresponds to importance sampling (Arulampalam et al., 2002) in a model where $p(\mathbf{y}_t|\mathcal{I}, l_t)$ is relaxed from a Dirac-delta distribution to a Gaussian. See the appendix for details.

## 5  TRAINING WITH PARTIAL SUPERVISION

The previous section describes how to annotate an image with a near-optimal sequence of glimpse locations for a particular image classification task. This section assumes that these, or other forms of glimpse sequence (e.g. the handcrafted glimpse sequences in Section 6), exist for all, or some, images in a dataset. These can then be used to partially supervise the training of a hard attention mechanism on this dataset. We refer to glimpse sequences used in this way as supervision sequences. We use separate losses for supervised (i.e. annotated with both a class label and a sequence of glimpse locations) and unsupervised (i.e. annotated with a class label but not glimpse locations) examples. By minimising the sum of these losses, our procedure can be viewed as maximising the joint log-likelihood of the class labels and supervision sequences. To be precise, let $q_\phi(\theta^i, l_{1:T}^i|\mathcal{I}^i)$ be a network's joint distribution over the chosen glimpse locations and predicted class label on image

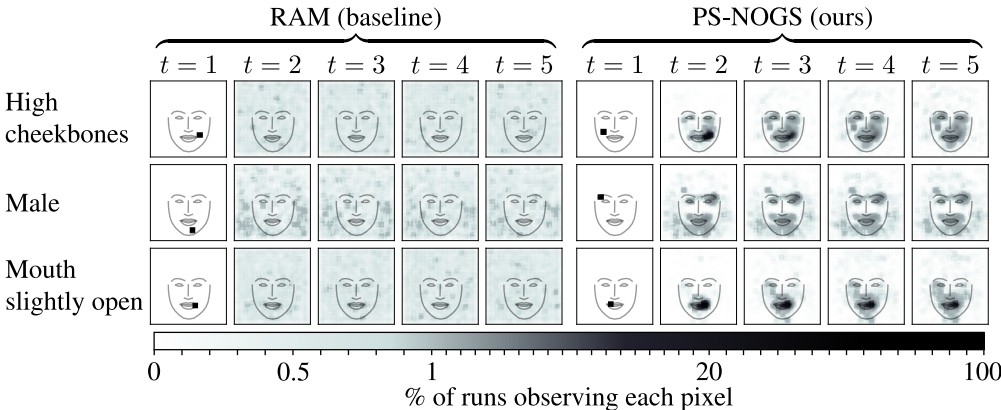

Figure 4: Comparison of glimpse locations chosen by RAM and PS-NOGS on the CelebA-HQ test set for three classification tasks. For each $t \in \{1, 2, 3, 4, 5\}$, we show an image where each pixel's colour corresponds to how often it was observed at this time step during testing. The outlines are produced by averaging outputs from a face detector across the dataset. For $t = 1$, each network learns a single location which it attends to on every test image. This is expected behaviour as the first location is chosen before taking any glimpses, and therefore before being able to condition on the image. RAM appears to then fail to learn to direct the later glimpses, attending almost uniformly across the image. In contrast, PS-NOGS distributes these glimpses broadly over the salient regions.

$\mathcal{I}^i$. Let $q_\phi(\theta^i | \mathcal{I}^i)$ be the marginalisation of this distribution over $l^i_{1:t}$. We maximise a lower bound on

$$\mathbf{L} = \sum_{i \in \text{sup.}} \overbrace{\log q_\phi(\theta^i, l^i_{1:T} | \mathcal{I}^i)}^{\text{supervised objective}} + \sum_{i \in \text{unsup.}} \overbrace{\log q_\phi(\theta^i | \mathcal{I}^i)}^{\text{unsupervised objective}} . \quad (8)$$

where 'sup' is the set of training indices with supervision sequences, and 'unsup' is the remainder. When running on unsupervised examples, we follow Mnih et al. (2014) and train the location network with a REINFORCE estimate of the gradient of the accuracy, using a learned baseline to reduce the variance of this estimate. Meanwhile, all other network components are trained to maximise the log-likelihood of the class labels (i.e. minimise a cross-entropy loss). Ba et al. (2014) noted that this maximises a lower bound on the unsupervised objective in Eq. (8). For examples with supervision sequences, the supervised objective in Eq. (8) is maximised by gradient backpropagation. The loss is computed by running the network with its glimpse locations fixed to those in the supervision sequence. The location network is updated to maximise the probability of outputting these locations while, as for unsupervised examples, the other network modules are trained to maximise the likelihood of the class labels. Minibatches can contain both supervised and unsupervised examples, with gradients computed simultaneuosly. We emphasise that supervision sequences are used throughout training. Although it is common to attenuate such supervision signals so that an end-to-end loss eventually dominates, preliminary experiments showed no improvements from this.

## 6 EXPERIMENTS AND RESULTS

**Datasets and network architectures**  We test our approach on CelebA-HQ (Karras et al., 2017) and a cropped variant of Caltech-UCSD Birds (CUB) (Wah et al., 2011). For both, GANs exist which satisfy the requirement to have a convincing generative model (Karras et al., 2018; Singh et al., 2019). The RNN is a GRU (Cho et al., 2014) and we use a simple classifier and location network architecture (see Appendix B for details). For both datasets, we use $T = 5$. The dataset-specific details are as follows: **(1) CelebA-HQ** Our experiments tackle 40 different binary classification tasks, corresponding to the 40 labelled attributes. We resize the images to $224 \times 224$ and use training, validation, and test sets of $27\,000$, $500$, and $2500$ images respectively. We use $16 \times 16$ pixel glimpses, with a $50 \times 50$ grid of allowed glimpse locations. The glimpse network has two convolutional layers followed by a linear layer. **(2) CUB** We perform 200-way classification of bird species. We crop the images using the provided bounding boxes and resize them to $128 \times 128$. Cropping is necessary because good generative models do not exist for the uncropped dataset, but there is still considerable

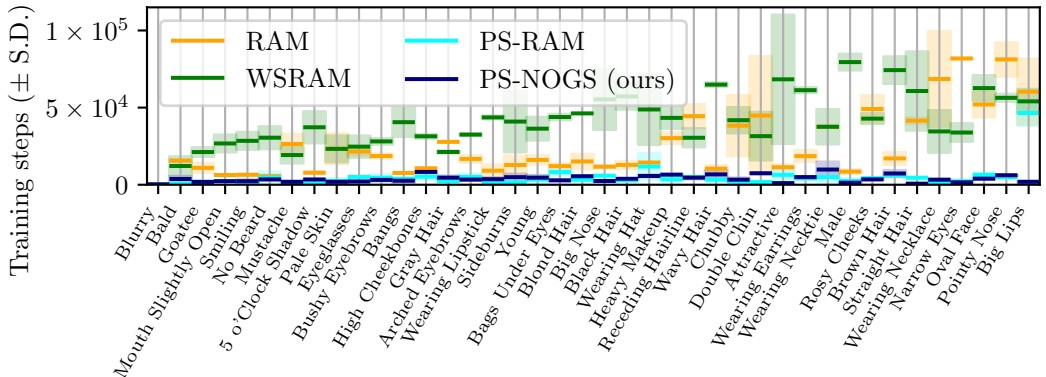

Figure 5: Number of training iterations for each CelebA-HQ attribute before a validation cross-entropy loss within 0.01 of the best is achieved. On average, PS-NOGS trains almost $7\times$ faster than RAM, the fastest method without supervision sequences. PS-NOGS also exhibits greatly reduced variance in the training time. Attributes are sorted by the mean training time.

variation in pose after cropping. We use 5120 training images, a validation set of 874 and a test set of 5751 (having removed 43 images also found in ImageNet). We use $32 \times 32$ pixel glimpses and a $12 \times 12$ grid of allowed glimpse locations so that adjacent locations are 8 pixels apart. The glimpse network is the first 12 convolutional layers of a VGG pretrained on ImageNet (Simonyan & Zisserman, 2014; Deng et al., 2009).

**BOED** We create 600 near-optimal glimpse sequences for each of the 40 CelebA-HQ classification tasks, and 1000 for CUB. This is a one-off computation that need only be done once for a particular task. It took 20 GPU-hours for CUB, and 10 GPU days for each CelebA-HQ task. We will publicly release these sequences along with our code, allowing them to be re-used by anyone to speed up the training of hard attention networks on these tasks.

**Baselines** All methods we compare use the same neural architecture; only the training algorithm is varied. For each experiment we compare against the RAM algorithm (Mnih et al., 2014), which is equivalent to the special case of our partially supervised objective with zero supervision sequences. We also compare against training with wake-sleep (WSRAM) (Ba et al., 2015), which we describe in Section 7. Furthermore, we consider training with supervision sequences created using two baseline methods. One is PS-RAM, which involves training a hard attention network with RAM and then sampling supervision sequences from the learned policy. The second, which we use on CUB, is to create "hand-crafted glimpse sequences" (PS-HGS). These are designed, using CUB's hand-annotated features, to attend to the beak, eye, forehead, belly and feet (in that order). If any of these are obscured, they are replaced by a

randomly selected visible body part. For both partially supervised baselines, we use the same number of supervision sequences as with PS-NOGS. We have not observed significant performance improvements from varying this number.

**Partial-supervision for CelebA-HQ** In Fig. 5, we plot the number of iterations until convergence on each CelebA-HQ classification task. On almost all tasks, the methods which involve partial supervision (PS-RAM and PS-NOGS) are faster than those which do not. Table 1 corroborates this finding, showing that PS-NOGS trains an average of $6.6\times$ faster on CelebA-HQ than RAM+, the fastest unsupervised method. Furthermore, the test accuracy for the partially supervised methods is competitive with the unsupervised baselines. Fig. 4 compares the learned attention policies for several tasks, with the re-

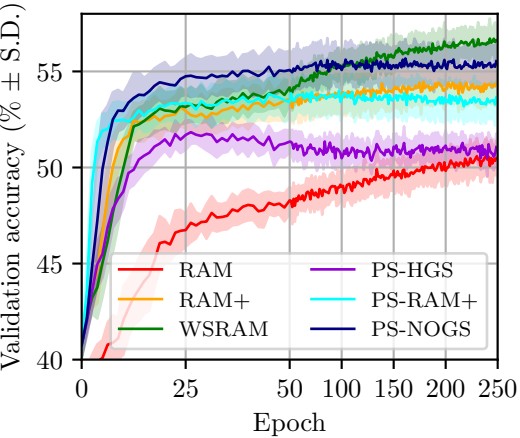

Figure 6: CUB validation accuracy over training.

Table 1: Results summary

| Method | CelebA-HQ (avg.) | | CUB | |
| | Iterations | Accuracy (%) | Iterations | Accuracy (%) |
| --- | --- | --- | --- | --- |
| RAM / RAM+ | 25 100 | 87.1 | 5690 | 53.7 |
| WSRAM | 40 700 | 88.1 | 11 800 | 55.7 |
| PS-HGS | - | - | 1790 | 50.5 |
| PS-RAM / PS-RAM+ | 4750 | 87.0 | 1980 | 53.2 |
| PS-NOGS (ours) | 3820 | 87.0 | 3530 | 54.4 |

mainder in the appendix. Unlike RAM, PS-NOGS appears to learn a suitable policy at every time step. These intuitively reasonable policies may explain why PS-NOGS trains 20% faster than PS-RAM.

**Partial-supervision for CUB**  On CUB, a pretraining stage is necessary to achieve high accuracy. We therefore pretrain the classifier, RNN, and glimpse network with glimpse locations sampled independently at each time step from either a uniform distribution (in the case of RAM) or from a heuristic which assigns higher probability to more salient locations, as estimated using the AVP-CNN (RAM+ and all others). See the appendix for details. Figure 6 shows the validation accuracy of each method throughout training. As seen on CelebA-HQ, all methods with partial supervision train faster than any without; Table 1 show the number of iterations on CUB before achieving a validation accuracy within 1% of the highest. Of the methods with partial supervision, PS-HGS quickly saturates and achieves the lowest test accuracy of the methods we compare. PS-RAM+ achieves slightly lower accuracy than RAM+, whose policy it is trained to imitate. PS-NOGS achieves the highest accuracy of the partially supervised methods, indicating that near-optimal glimpse sequences are more suitable for training than any of the supervision sequences used by our baselines. However, we note that after sufficiently many training epochs (approximately 100), WSRAM achieves higher validation accuracy than PS-NOGS, and correspondingly achieves higher test accuracy. This may be due to some bias introduced by approximations in the generation of near-optimal glimpse sequences, which will not go away with further training.

# 7 RELATED WORK

**Variational approaches**  A notable body of work (Ba et al., 2015; Lawson et al., 2018; Shankar & Sarawagi, 2018) frames the glimpse locations as latent variables and trains an inference network to approximate $q_\phi(l_{1:T}|\theta, \mathcal{I})$. This allows for better estimates of the objective $\log q_\phi(\theta|\mathcal{I})$, which is marginalised over glimpse locations. Our comparison against WSRAM (Ba et al., 2015) indicates that using supervision sequences allows faster training than these variational techniques alone. Further, supervision sequences could be used in conjunction with these techniques (Teng et al., 2020).

**Hard attention architectures**  Elsayed et al. (2019) recently demonstrated a hard attention network which achieved accuracy on ImageNet (Deng et al., 2009) close to that of CNNs which use the whole image. However, their approach neccesitates running a convolutional network on the entire image to select glimpse locations. As such, they advertise improvements in interpretability rather than computational efficiency. Sermanet et al. (2014) train a hard attention architecture with REINFORCE to achieve state-of-the-art accuracy on the Stanford Dogs dataset. In addition to accessing the full image in low resolution at the start, they use large glimpses (multiple $96 \times 96$ pixel patches at different resolutions) to effectively solve the task in a single step. This avoids problems resulting from learning long sequences with REINFORCE but also rules out the computational gains possible with smaller glimpses. Katharopoulos & Fleuret (2019) proposed a form of hard attention where, after processing the downsampled image, multiple glimpses are sampled and processed simultaneuosly. This is again incompatible with a low-power setting where we cannot afford to operate on the full image.

**Supervised attention**  We are not alone in providing supervision targets for an attention mechanism. This is common for soft attention in visual question answering, where targets have been created by human subjects, either with gaze-tracking (Yu et al., 2017) or explicit annotation (Das et al., 2017). Either way is expensive and dataset-specific (Das et al., 2017; Gan et al., 2017). Recent work has reduced this cost by e.g. extrapolating supervision signals from annotated datasets to other, related,

datasets (Qiao et al., 2018), or using existing segmentations to speed up annotation (Gan et al., 2017). Even so, considerable human effort is required. PS-NOGS automates this for image classification.

## 8 DISCUSSION AND CONCLUSION

We have demonstrated a novel BOED pipeline for generating near-optimal sequences of glimpse locations. We also introduced a partially supervised training objective which uses such a supervision signal to speed up the training of a hard attention mechanism. By investing up-front computation in creating near-optimal glimpse sequences for supervision, this speed up can be achieved along with comparable final accuracy. Since we release the near-optimal glimpse sequences we generated, faster training and experimentation on these tasks is available to the public without the cost of generating new sequences. Our work could also have applications in neural architecture search, where this cost can be amortised over as many as 12 800 (Zoph & Le, 2016) training runs with different architectures. Finally, our framework could also be extended to attention tasks such as question answering where the latent variable of interest is richly structured, or scaled to more complex images with structured image completion models (Dai et al., 2018).

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
