# OpenReview forum: "Near-Optimal Glimpse Sequences for Training Hard Attention Neural Networks"
_ICLR.cc/2021/Conference — Reject_

### Official Review · AnonReviewer4 · 2020-10-23
**Complicated and not optimal**

**Rating:** 4
**Confidence:** 4

**Review:**

The paper trains hard attention for image classification. The network is partially supervised by attention locations proposed to maximally reduce the entropy of the image label distribution. To propose these locations, the method needs an already trained image classifier conditioned on glimpses and their locations. Additionally, the method needs a generator of images, conditioned on the glimpses and their locations. This generator is approximated by searching a set of 1.5 million pre-generated images for close matches.

It is visible that the paper required a lot of work. Unfortunately, the method is not very practical and the method is not better than simpler alternatives.

Pros:
- The paper is well written.

Main cons:
1) The greedy minimization of the expected posterior entropy is not optimal, if additional glimpses can be done in the next steps.
2) The searching in the pre-generated dataset of images is not practical. It would work worse on more diverse images. The method was tested only on faces and birds.
3) If the aim is to speed-up training of new networks, a simpler alternative is to distill the attention locations and classification outputs from a pretrained Recurrent Attention Model (RAM+).

---

> ### Author Response · Authors · 2020-11-13
> **Thanks for your review and comments**
>
> Thank you for your review and thoughtful comments. We will respond to your first two issues now, and respond to the third after running experiments:
>
> **“greedy minimization of the expected posterior entropy is not optimal, if additional glimpses can be done in the next steps.”** Indeed. This is an extremely common heuristic in the Bayesian experimental design literature, and is part of the reason we describe our method as “near-optimal” rather than “optimal”.
>
> **The searching in the pre-generated dataset of images is not practical. It would work worse on more diverse images.** In the section “Stochastic image completion”, we acknowledge these drawbacks but note that it is still the best of the many methods we tested. As you mention, we demonstrate our method on two very different datasets despite the limitations our image completion module imposes. Future improvements in, e.g., conditional GANs are likely to improve the scalability of our method significantly.

---

> > ### Comment · AnonReviewer4 · 2020-11-23
> > **Response to authors**
> >
> > Thank you for the response.
> > I hope you will have the extra distillation experiments finished before the end of the discussion period.

---

> > > ### Author Response · Authors · 2020-11-23
> > > **Updated paper with new baseline**
> > >
> > > Thank you for this suggestion (and your patience). We have updated the paper with a new “PS-RAM” baseline. Based on your suggestion, it takes a network trained with RAM and uses it to generate supervision sequences. We then train later networks using our partially supervised training objective with these supervision sequences. To summarise the results, we find that PS-RAM leads to faster training than RAM with a small cost to accuracy (0.1% on average for CelebA-HQ and 0.5% for CUB). Comparing PS-RAM to PS-NOGS, we see that they reach the same accuracy on CelebA-HQ but PS-NOGS trains 20% faster. On CUB, we see that PS-NOGS achieves a final accuracy 1.2% higher. The difference in performance may be explained by the effect shown in Figure 4: glimpses sampled with RAM are almost uniformly distributed after the first time step, whereas glimpses made with PS-NOGS are concentrated on the salient regions and so may provide a better learning signal

---

> > > > ### Comment · AnonReviewer4 · 2020-11-23
> > > > **Response to authors**
> > > >
> > > > Thank you for the honest PS-RAM(+) experiments.

---

### Official Review · AnonReviewer1 · 2020-10-25
**not well motivated**

**Rating:** 5
**Confidence:** 4

**Review:**

This paper introduces a way to annotate a glimpse sequence for an image. It uses BAYESIAN OPTIMAL EXPERIMENTAL DESIGN to achieve this. Using the obtained annotations, hard attention can be trained with a partially supervised way.

My concerns mainly focus on the motivation and evaluation. Specifically, the authors claim using hard attention to do image classification may enable the use of modern approaches to computer vision in low-power settings such as mobile devices. But from the paper (besides the computation of glimpse sequence is computationally expensive.) at the end, the proposed method needs to run the glimpse network several times, and the structure of the glimpse network is similar to standard classification network, which needs only a single forward pass on the whole image. I'm doubt adopting the proposed approach can truly lead to a low power solution.  Image classification seems like an unsuitable task to show the benefits of hard attention.  Even if we would like to test on image classification, we need to include more baselines, such as standard convnets and compare them to the proposed method in terms of computational complexity and effectiveness.

Other issues:
  Stochastic image completion involves creating an empirical image distribution with 1.5 million images. Will this lead to an unfair comparison or data leakage?
  It's better to include standard datasets for image classification such as ImageNet, CIFAR, stc.
  It's interesting to see whether a hard attention method trained with REINFORCE will obtain a similar glimpse sequence and the proposed annotation method.

---

> ### Author Response · Authors · 2020-11-13
> **Thank you for your thoughtful comments**
>
> Thank you for your review and thoughtful consideration of the paper’s motivation.
>
> **Not suitable for low-power settings as the structure of the glimpse network is similar to a standard classification network** The structure is indeed similar, but vastly less computation is required: the number of FLOPs required for a convolution operation scales with the number of input pixels, and so is greatly reduced when only a fraction of the image is processed.
>
> **Data leakage from empirical distribution** I believe you are referring to the issue which we briefly address on page 4 (“We note that the use of pre-trained models makes test leakage possible but verify in Appendix C.4 that this is unlikely to impact our results.”). The results presented in Appendix C.4 lead us to believe that this is unlikely to significantly affect our results.
>
> **“It's better to include standard datasets for image classification such as ImageNet, CIFAR, stc”** We appreciate that this would be ideal. However, as mentioned on page 6, we are limited to using datasets for which we have convincing generative models (and convincing conditional generative models for image completion). We believe that future advances in conditional generative modelling will allow our method to scale to more complex datasets such as ImageNet.

---

> > ### Comment · AnonReviewer1 · 2020-11-24
> > **Thank you for your response**
> >
> > At first thanks for your detailed response, which solves my concerns in terms of data leakage.
> > However, I have few more questions:
> >
> > 1. I think  the paper could be clearer about whether the focus of this paper is obtaining gt glimpse sequences as *annotations* or a *method* to do hard attention.
> > 2. In terms of low power setting, although the FLOPs of a single run is lower, the proposed method contains multiple runs, while standard CNNs require only one run. It's better if there could be a comparison of resource costs or time costs. Moreover, a standard CNN could be included as a baseline to demonstrate the trade-off between accuracy and power.
> > 3. it's better if you could include some observation on found glimpse sequences, such as the common patterns in these sequences.

---

> > > ### Author Response · Authors · 2020-11-25
> > > **Preliminary comparison with CNNs in appendix**
> > >
> > > Thanks again for your engagement and further comments.
> > >
> > > 1. We view the primary contribution of this work as a method for speeding up hard attention training. Although this involves initially annotating the data with glimpse sequences, the primary purpose of these is to use in our method for faster training. We will ensure this is clear in the paper.
> > >
> > > 2. First, we would like to clarify that we only require multiple runs in that the glimpse network is used at each timestep 1,...,T. We run the hard attention network (for T steps) only once to classify each image. This is in contrast to e.g. “Multiple Object Recognition with Visual Attention” (Ba et al., 2015), who average classification outputs over multiple runs of a hard attention network. We will edit the manuscript to state this explicitly. Regarding the comparison with a standard CNN, we have updated the appendix with preliminary results of such a comparison (see Appendix D). This shows that, by using a form of “early stopping”, where the network can choose to take less than $T$ glimpses, hard attention can outperform state-of-the-art CNNs (at least for less than $7 \times 10^5$ FLOPs). Before the camera-ready, we will provide a more systematic study and investigate whether different hard attention architectures can provide better performance for regimes where more FLOPs are allowed.
> > >
> > > 3. We show examples of near-optimal glimpse sequences for CUB in Figure 13. For CelebA-HQ, we show heatmaps of glimpse locations trained with PS-NOGS in Figures 4 and 9-11. Although these are not exactly the near-optimal glimpse sequences, they follow a very similar distribution. We have not observed any patterns of note that are not visible in these figures, or described in the captions.

---

### Official Review · AnonReviewer2 · 2020-10-30
**A well thought out procedure to learn a hard attention mechanism over real images**

**Rating:** 6
**Confidence:** 2

**Review:**

This paper presents a learning framework for a hard attention mechanism. The glimpses captured by the attention mechanism are guided by the goal of minimizing output uncertainty for a downstream task such as classification. The authors pose this problem in a probabilistic framework which is based on Bayesian optimal experimental design (BOED). They devise a tractable approximation to the entropy over images and glimpse sequences and search for the glimpse sequences which minimize the output entropy of a recurrent classification model.

With the ability to incorporate supervised glimpse training into the attention mechanism, the authors are able to alleviate the cold start issue which many hard attention models over images suffer from.

While the probabilistic formulation described in the paper is fairly detailed, a figure showing how the attentional variational posterior ($g_{AVP}$) is integrated into the rest of the attention model in Figure 1. Is $g_{AVP}$ trained through some form of model distillation with the attention model?

What types of hand-crafted glimpse sequences are beneficial for the model? CUBs annotated features were not intended to be a glimpse sequence, so how was the order of the annotations chosen?

Can the authors provide a pseudo-code description of the attention algorithm? This would help clarify the pipeline of how $g_{AVP}$ is both trained and integrated into the larger model.

---

> ### Author Response · Authors · 2020-11-13
> **Thanks for the helpful comments and questions**
>
> Thank you for taking the time to review our paper and leave these thoughtful comments.
>
> **Clarification of integration of $g_{AVP}$ with the rest of the model (and suggestion for pseudocode)** Thank you for drawing our attention to this unclear point. $g_{AVP}$ is used only for creating near-optimal glimpse sequences in the BOED phase. It is not used with the hard attention model while performing semi-supervised training, nor at test time. We have updated the appendix with more pseudocode (Algorithm 1) as you suggest.
>
> **“What types of hand-crafted glimpse sequences are beneficial for the model? […] how was the order of the annotations chosen?”** There is no clearly best ordering of glimpse locations for CUB. We chose the parts to attend to, and order in which to attend to them, based on what seems most useful for a human attempting to identify the bird species. We believe this gives a reasonable baseline, although it is by no means optimal.

---

### Official Review · AnonReviewer3 · 2020-11-02
**Novel idea of learning hard visual attention using Bayesian Optimal Experimental Design**

**Rating:** 7
**Confidence:** 2

**Review:**

This paper frames hard visual attention as a Bayesian optimal experimental design (BOED) problem from which the optimal locations to attend to are those with greatest expected reduction in the entropy of classification. With the methodology from BOED literature, this paper approximate the optimal behavior to generate 'near optimal' sequences to partially supervise the learning of hard attention.
Overall, I think the paper is easy to read, the proposed idea is solid and well-motivated, although the actual implementation seems to be quite complicated which involves additional two different models (attentional variational posterior and image completion model). It is hard to tell how the learning of these modules will affect the final performance.

Additionally I have some specific questions regarding to the paper:
(1) I am a bit confused with Figure 3 and Sec 4. It is mentioned that the mask is concatenated as an additional channel. However, since you already mask the unobserved pixels to zero, why it is still needed?
(2) Following Eq (7), is $g_{AVP}$ trained with randomly sampled glimpses? How can we make sure we can predict the label correctly from a random glimpse? Also, why it is called "attentional" posterior?
(3) How the quality of $r_{img}$ will impact the results? Do we need to train a different $r_{img}$ first every time before testing on a new dataset? Is it possible to only condition on the current image to create glimpse supervisions with $g_{AVP}$ instead of using image completion?
(4) How to inspect if the generated supervisions might hurt the performance? For instance, as the supervisions are generated to approximately maximize the entropy reduction, it is still possible that such attention is not optimal for neural network to predict the final label.

---

> ### Author Response · Authors · 2020-11-13
> **Thanks for the helpful feedback**
>
> Thank you for your time and helpful comments.  In answer to your questions:
>
> **Why is the mask concatenated as an additional channel (Fig. 3)?** Consider the case where an observed patch is grey and featureless, such that the normalised pixel values are all zero. Concatenating the mask would then be necessary to know that this was observed and grey, rather than simply unobserved. While it is unclear whether this would be an issue in practice, concatenating the mask prevents any such problems for very little cost.
>
> **Is $g_{AVP}$ trained with random glimpses?** The glimpses are random in the sense that their locations are sampled randomly. This will mean that they are sometimes taken in an “uninformative” location and the label cannot be accurately guessed. This is not a problem: we merely need $g_{AVP}$, to approximate the posterior over class labels given a series of glimpses; it does not need to “correctly” guess the class label every time.
>
> **Why do we name it the attentional variational posterior?** We use “attentional” to emphasise that the distribution is conditioned on the parts of the images that have been attended to and not, for example, the whole image.
>
> **Do we need to train a different $r_{img}$ every time before testing on a new dataset? Is it possible to only condition on the current image to create glimpse supervisions with instead of using image completion?** We do indeed need to create $r_{img}$ separately for every dataset. The method you suggest of using the current image would mean estimating the expectation in Eq. 5 with only a single sample of $\mathcal{I}$, so is unlikely to work well.
>
> **Can we if tell the generated supervision sequences might hurt the performance?** We can only suggest training both with and without supervision, and comparing the results, although this is an interesting avenue for future work.

---

### Author Response · Authors · 2020-11-23
**Updated experiments**

We have uploaded a new version of the paper with updated experiments. There are several changes:

- The primary change is that, as suggested by AnonReviewer4, we include a baseline where supervision sequences are sampled from a trained RAM network’s policy.

- We have also added the WS-RAM baseline (previously shown only on CelebA-HQ) for CUB.

- Additionally, the experiments previously shown for CUB initialised all models from the same pretrained network (following the pretraining procedure described in Section 6). We now repeat the experiments with 5 different pretrained initializations (obtained with different random seeds) and average the results.

---

### Decision · Program_Chairs · 2021-01-07
**Final Decision**

**Decision:**

Reject

**Comment:**

This paper proposes a new and unusual way of training hard attention mechanisms in vision models. Instead of training with reinforcement learning (as is typical), the authors develop a procedure for generating "glimpse sequences" that can be effectively used as supervision. Models trained in this way produce qualitatively "better" glimpse sequences, higher accuracy, and converge in fewer steps. There was some disagreement and discussion about the merits of the paper. Overall, there were some major concerns:
- The process for obtaining glimpse sequences is very computationally expensive. The authors argue that this cost can be amortized because the same glimpse sequences can be computed once for a given dataset and reused. However, there was limited real-world motivation for this setup, apart from mentioning neural architecture search (a niche method that is not widely used, and probably has never been used to develop hard-attention models).
- The method relies on a "convincing" generative model for a given dataset. This limits experiments to simple datasets with unrealistically-constrained visual structure. The authors point out that as generative models get better, their method could be applied to more realistic datasets, but as it stands the experimental validation is correspondingly weak.
- The improvement in performance is not huge - the "WSRAM" baseline outperforms the use of "near-optimal" glimpse sequences. While it is certainly true that the proposed method converges faster, the fact that the proposed method requires such an expensive preprocessing step downgrades this benefit significantly
- While the authors provided insightful distillation-based baselines in the rebuttal, it remains to be seen whether simple distillation from stronger RAM models (e.g. WSRAM which outperforms the proposed method) could be made to work better/more efficiently.
These factors lead to a reject decision overall.